Interaction of CTCF and CTCFL in genome regulation through chromatin architecture during the spermatogenesis and carcinogenesis

Tong Xin
Gao Yang gaoyang@stu.edu.cn
Su Zhongjing g_zjsu@stu.edu.cn
Department of Histology and Embryology, Shantou University Medical College , Shantou , Guangdong , China
Uversky Vladimir
Electronic publication date: 2024 Oct 15
Publication date: 2024
Volume: 12
Electronic Location ID: e18240
Received 2024 Jul 17; Accepted 2024 Sep 15
Copyright: ©2024 Tong et al.
Copyright year: 2024
Copyright holder: Tong et al.
License: This is an open access article distributed under the terms of the Creative Commons Attribution License, which permits unrestricted use, distribution, reproduction and adaptation in any medium and for any purpose provided that it is properly attributed. For attribution, the original author(s), title, publication source (PeerJ) and either DOI or URL of the article must be cited.
License URL: https://creativecommons.org/licenses/by/4.0/

Keywords: CTCF, CTCFL, Chromatin architecture, Spermatogenesis, Carcinogenesis

Funding: National Natural Science Foundation of China No. 31870745 and 32201016 This work was supported by the National Natural Science Foundation of China (No. 31870745 and 32201016). The funders had no role in study design, data collection and analysis, decision to publish, or preparation of the manuscript.

==============================
The zinc finger protein CTCF is ubiquitously expressed and is integral to the regulation of chromatin architecture through its interaction with cohesin. Conversely, CTCFL expression is predominantly restricted to the adult male testis but is aberrantly expressed in certain cancers. Despite their distinct expression patterns, the cooperative and competitive mechanisms by which CTCF and CTCFL regulate target gene expression in spermatocytes and cancer cells remain inadequately understood. In this review, we comprehensively examine the literature on the divergent amino acid sequences, target sites, expression profiles and functions of CTCF and CTCFL in normal tissues and cancers. We further elucidate the mechanisms by which CTCFL competitively or cooperatively binds to CTCF target sites during spermatogenesis and carcinogenesis to modulate chromatin architecture. We mainly focus on the role of CTCFL in testicular and cancer development, highlighting its interaction with CTCF at CTCF binding sites to regulate target genes. In the testis, CTCF and CTCFL cooperate to regulate the expression of testis-specific genes, essential for proper germ cell progression. In cancers, CTCFL overexpression competes with CTCF for DNA binding, leading to aberrant gene expression, a more relaxed chromatin state, and altered chromatin loops. By uncovering the roles of CTCF and CTCFL in spermatogenesis and carcinogenesis, we can better understand the implications of aberrant CTCFL expression in altering chromatin loops and its contribution to disease pathogenesis.

Introduction

CTCF (CCCTC-binding factor) was originally described as a unique “multivalent transcriptional factor” to transcriptionally repress the c-myc oncogene (Filippova et al., 1996; Klenova et al., 1993; Lobanenkov et al., 1990). CTCF is a highly conserved zinc finger protein that plays a critical role in organizing the genome into distinct functional units for gene expression (Phillips & Corces, 2009). CTCF binds to DNA at specific sites known as CTCF-binding sites, which often demarcate boundaries between chromatin domains. CTCFL (CTCF like), also known as BORIS (Brother of the Regulator of Imprinted Sites), was initially identified in the male testis as a paralogue of CTCF, with a role in epigenetic reprogramming and regulation of the testis-specific gene expression (Kosaka-Suzuki et al., 2011; Loukinov et al., 2002; Suzuki et al., 2010). Recent studies have shown that CTCF and CTCFL are co-expressed at all stages of spermatogenesis and in certain cancer cells (Rivero-Hinojosa et al., 2021), suggesting their involvement in both the physiological process of male gametogenesis and tumorigenesis to modulate chromatin architecture.

The three-dimensional (3D) topology of chromatin including chromosome territories (CTs), compartments, topologically associating domains (TADs) and chromatin loops elucidates how two meters of DNA can be compactly organized within the nucleus, which is only a few micrometers in diameter (Deng, Feng & Pauklin, 2022; Lieberman-Aiden et al., 2009; Matharu & Ahituv, 2015; Su et al., 2020). At the first level of chromatin structure is organized around nucleosomes, where DNA is wrapped around a histone octamer (Khorasanizadeh, 2004). These 3D architectures play crucial roles in biological processes, and disruptions in these structures can lead to aberrant gene expression and cancer development. Recent studies have extended the relevance of TADs and their boundaries beyond cancer to include immune, inflammatory, reproductive, and other rare disease pathogenesis (Deng, Feng & Pauklin, 2022; Hanahan, 2022). Chromatin loops, mediated by CTCF in combination with cohesin, bring promoters and enhancers into proper spatial distance and maintain the transcriptional levels (Rao et al., 2014). CTCF and CTCFL, due to their conserved 11-zinc finger (ZF) DNA-binding domain, can bind to the same DNA sequences. However, the mechanisms by which CTCF and CTCFL interact to regulate the gene transcription in the testis and cancer, and whether they alter 3D chromatin architecture, remain poorly understood.

This review aims to provide a comprehensive analysis of the distinct amino acid sequences, the target sites, expressions patterns and functions of CTCF and CTCFL in normal tissues and cancers. We will discuss the mechanisms by which CTCFL may competitively or cooperatively interact with CTCF to modulate chromatin architecture during spermatogenesis and carcinogenesis. This review represents the first effort to systematically summarize how differential expression patterns of CTCF and CTCFL influence gene regulation and chromatin architecture across normal somatic cells, sperm and cancer cells.

Survey Methodology

A comprehensive literature search was performed using the PubMed databases, focusing primarily on articles published within the last five years. However, earlier seminal studies were also included due to their foundational contributions. The search utilized the following keywords: CTCF, CTCFL, spermatogenesis, cancer and chromatin architecture. Both original research articles and review papers were considered if they provided insights into the distinctions between CTCF and CTCFL, including the structural characterization, their interactions with each other and with cohesin in shaping chromatin architecture, the similarity and difference in expression patterns and functions, and their regulatory roles in gene expression within sperm and cancer contexts. This review is intended to serve researchers and clinicians interested in understanding the relationship between CTCF and CTCFL in regulating testis and cancer-related genes, as well as the implications of 3D chromatin architecture alterations in these tissues.

The sequence difference between CTCF and CTCFL

CTCF, a ubiquitously expressed protein, is highly conserved across species, with approximately 90% amino acid identity from chickens to higher mammals, including humans (Klenova et al., 2002; Klenova et al., 1993; Lobanenkov et al., 1990). CTCFL retains the conserved zinc finger domain in humans and mice but is absent in chickens, which illustrated that CTCFL is subsequently divergent from CTCF sequence during vertebrate evolution. Both CTCF and CTCFL possess an 11-zinc finger (ZF) DNA-binding domain, though their N- and C-terminal regions exhibit significant divergence, as depicted in Fig. 1A. The N- and C-terminal regions of CTCFL differ significantly from those of CTCF, with only a 15% sequence overlap. CTCFL’s 11-ZFs share high sequence homology with those of CTCF (74.1%), with the most significant differences observed in the ZF7 domain for the central ZFs 4–7 domains. The 3D structures of CTCF and CTCFL are shown in Fig. 1B. The N- and C-terminal structures are quite different between CTCF and CTCFL. CTCF can recognize between 50,000 to 80,000 distinct DNA sequences through its 11 ZFs in humans and mice, with these sequences categorized into the upstream, the core and the downstream motifs. Chromatin immunoprecipitation combined with tiling arrays (ChIP-seq) and systematic mutagenesis of CTCF ZFs have shown that ZFs 4–7 bind to approximately 80% of CTCF binding sites, which contain a 15–20 bp core motif. Around 13% of these sites include a 10 bp upstream motif, bound by ZFs 1–2 and separated from the core motif by a 5–6 nucleotide spacer. Additionally, 8% of the sites contain a 10 bp downstream motif, bound by ZFs 9–11 and separated from the core motif by a 6–8 nucleotide spacer (Nakahashi et al., 2013). Notably, upstream and downstream motifs can modulate CTCF binding in vivo, with the core motif being stabilized by interactions with these motifs (Marshall, Bailey & Rasko, 2014). However, the upstream and downstream motifs in CTCFL remain unclear. Specifically, the core motifs of CTCF and CTCFL differ in the absence of adenine at position 5 and a lower prevalence of adenine at position 8 in CTCFL, alongside a higher prevalence of guanine at positions 10 and 13 in the CTCFL motif relative to the CTCF (Sleutels et al., 2012). The DNA-binding motifs of CTCF and CTCFL are illustrated in Fig. 1C.

Figure 1 The sequences, 3D structures and motifs of CTCF and CTCFL.

(A) Alignment of the amino acids of CTCF and CTCFL in the human genome. The 11-ZF DNA binding domains of CTCF and CTCFL are highlighted in black box and numbered and the N-and C-terminal regions are marked. The conserved amino acids are shown with an asterisk (*); the colon (:) represents a strongly conservative amino acid substitution, and the period (.) represents a weakly conservative amino acid substitution. The alignment was generated with Clustal Omega. The underlined residues (black) represent a predicted nuclear export signal (NES) of CTCFL but is absent in CTCF. (B) The 3D structures of CTCF and CTCFL were generated with SWISS-MODEL using the AlphaFold model. The N-and C-terminal regions are marked. The zinc finger regions are highlighted in the 3D structures. (C) CTCF and CTCFL consensus motif found in CTCF-only and CTCFL-only occupied sites. Analysis of CTCF and CTCFL consensus motif using MEME discovery software, which identifies three distinct motifs: upstream (U), core (C), and downstream (D) DNA conserved element. Absolute number of peaks and distances are provided.

Initial studies indicated that the C-terminal region, combination with ZF11, plays a crucial role in the target search and self-association of CTCF. The C-terminal domain of CTCF is essential for binding the SA2 subunit of cohesin, thereby promoting cohesin occupancy and chromatin loop formation (Xiao, Wallace & Felsenfeld, 2011). Recent studies have demonstrated that a 79-amino acid segment within the N terminus of CTCF and the first two CTCF zinc fingers were essential for cohesin retention at CTCF binding sites and the maintenance of 3D chromatin conformation (Pugacheva et al., 2020). Additionally, the N-terminal region of CTCF is known to interact directly with the SA2-SCC1 subcomplex of cohesin, facilitating chromatin loop formation (Li et al., 2020). Both the N- and C-terminal regions of CTCF are essential for the cohesin retention. However, CTCFL is unable to bind cohesin due to the differences in its N- or C-terminal amino acid sequence. A chimeric protein, created by replacing the N-terminal region and the first two ZFs of CTCFL with those of CTCF, demonstrated enhanced cohesin retention compared to CTCFL, although it remained less effective than CTCF. These findings suggest that the 3D conformation of the CTCF-DNA complex contributes to spatial constraint on cohesin movement rather than the N- or C-terminal amino acid sequences. The N- and C-terminal regions of CTCFL differ significantly from those of CTCF contributes to their distinct functional roles (Marshall, Bailey & Rasko, 2014). The C-terminal regions of CTCFL contains putative nuclear export signal (Ogunkolade et al., 2013), seen in Fig. 1A. Consequently, CTCFL also functions as an RNA-binding protein, associated with actively translating ribosomes, and is involved in regulating gene expression at both the transcriptional and post-transcriptional levels. CTCFL is expressed as 23 different isoforms, encoding 17 distinct proteins, which are categorized into six subfamilies based on variations in their promoters, zinc-fingers, N- and C-terminal regions (Pugacheva et al., 2010). These alternative isoforms are found in both germ cells and cancers suggesting their potential unique functional significance. Using 5′RACE, three alternative promoters for CTCFL have been identified: including A (−1,447 bp), B (−899 bp) and C (−658 bp). In testicular and cancer cells, CTCFL is transcribed from these three promoters (Renaud et al., 2007). The regulation of CTCFL activity is complex, involving different promoters and cell types, indicating its possible influence on various aspects of spermatogenesis and carcinogenesis. Thus, while the zinc finger DNA-binding domains of CTCF and CTCFL enable binding to similar DNA sites, the differences in their 3D structures underlie their distinct functions.

The binding specificity of CTCF and CTCFL

The zinc finger domains of both CTCF and CTCFL determine their specific DNA-binding motifs. Although the DNA-binding sequences of CTCF and CTCFL exhibit a high degree of similarity, allowing both proteins to bind to the same DNA sequences in vivo, they interact with distinct protein partners, CTCFL associated with meiotic-specific cohesins, rather than CTCF binding cohesins (Boukaba et al., 2022). Notably, approximately two-third of CTCFL binding sites overlap with those of CTCF; however, the majority of CTCF binding sites do not overlap with CTCFL sites. Although the central 11-ZF DNA-binding domains of CTCF and CTCFL are highly conserved at the amino acid level, there are subtle differences in their amino acid sequences. The N- and C-terminal domains of these proteins influence their binding preferences, with CTCF primarily associating with intergenic and intronic regions, often acting distally from transcription start sites (TSSs) (Bergmaier et al., 2018). In contrast, CTCFL tends to bind closer to TSSs and associates with transcriptionally active genes, often overlapping with histone transcriptionally active modification (Marshall, Bailey & Rasko, 2014; Sleutels et al., 2012). CTCFL is predominantly located at active promoters and enhancers rather than CTCF located at intronic or intergenic regions (Nishana et al., 2020). Recent studies indicate that CTCFL can also bind to the CTCF binding sites at intronic or intergenic regions that have been epigenetically reprogrammed into alternative transcriptional start sites (Pugacheva et al., 2024). These observations suggest that CTCFL plays a more direct role in transcriptional activation compared to CTCF, likely through its involvement in epigenetic regulation.

CTCF target sites can be classified into single CTCF target sites (termed 1 ×CTSes) and sites containing two adjacent CTCF motifs (termed 2 ×CTSes), which may regulate divergent chromatin structures (Pugacheva et al., 2015). The 2 ×CTSes are indicative of two potential binding scenarios: either one molecule of CTCF and one molecule of CTCFL, or two molecules of CTCFL in CTCFL-positive cells, and two molecules of CTCF in CTCFL-negative cells. 2 ×CTSes are more prevalent than 1 ×CTSes in vivo. Simultaneous occupancy of CTCF and CTCFL at 2 ×CTSes has been confirmed in K562 and lymphoid Delta47 cells as well as in mouse round spermatids (Lobanenkov & Zentner, 2017; Sleutels et al., 2012). CTCFL preferentially binds to CTCF binding sites at 2 ×CTSes. Bergmaier et al. suggested that CTCFL, which binds less strongly, may require the presence of an additional CTCF molecule nearby to stabilize its binding (Bergmaier et al., 2018; Pugacheva et al., 2015). When CTCF and CTCFL are co-expressed in testis and cancer cells, they tend to form a heterodimer at 2×CTSes, suggesting a similar regulatory mechanism for target genes in these contexts. Additionally, CTCF binding is inhibited by DNA methylation (Phillips & Corces, 2009), while CTCFL preferentially binds to the methylated DNA (Singh et al., 2017). Thus, the competitive or cooperative binding of CTCFL to CTCF binding sites is associated with the regulation of target genes and site-specific DNA-methylation patterns.

The expression patterns of CTCF and CTCFL

Initially, the expression and distribution between CTCF and CTCFL in testis were reported in mutually exclusion, with CTCF presented in round spermatids (after meiosis) and CTCFL in primary spermatocytes (before meiosis and during meiotic prophase) (Loukinov et al., 2002). The recent study showed that CTCF and CTCFL were co-expressed all stages of spermatogenesis both in mice and humans through single-cell RNA-seq analysis. The results showed that CTCF expression covered all stages of spermatogenesis but CTCFL expression was highest in spermatogonia, decreased in spermatocytes, and expressed again in early round spermatids, adding complexity to the localization and expression of CTCFL (Rivero-Hinojosa et al., 2021). In the testis, CTCFL expression is associated with the genome-wide demethylation (Loukinov et al., 2002; Renaud et al., 2007). This genome-wide DNA demethylation regulates meiotic recombination and may link CTCFL-CTCF switching to the initiation of de novo DNA-methylation. De novo DNA-methylation appears to involve histone H3 lysine methylases, though no direct role of CTCFL or CTCF in histone methylation and DNA methylation has been established (Klenova et al., 2002). Besides the expression in male spermatogenesis, CTCFL was also reported in certain cancers. Typically, a transcriptional inhibitor, CTCF binds to the promoter of CTCFL and transcriptionally suppresses its expression in somatic cells. Moreover, p53-mediated DNA methylation also represses CTCFL expression in somatic cells (Park et al., 2004; Renaud et al., 2007). Thus, CTCFL expression is predominantly restricted to the adult male testis. In certain cancer cell lines, p53 deletions or mutations can lead to the re-expression of CTCFL. Another mechanism leading to re-expression of CTCFL in cancer is epigenetic promoter hypomethylation. CTCFL expression is closely associated with the promoter methylation and the overall genome-wide methylation state.

Unlike CTCFL transcriptionally activated, CTCF is more frequently mutant in cancers, including the mutations in CTCF-binding motifs (Debaugny & Skok, 2020). Pan-cancer analysis of whole genomes (ICGC/TCGA database) has revealed that the mutations, deletions and amplifications of CTCF are prevalent in mature B-cell lymphoma, bladder cancer, uterine endometrioid carcinoma, ovarian cancer. The functional enrichment of co-occurring mutant genes analyzed with the Metascape platform show that the enrichment pathway is focus on chromatin organization, DNA metabolic process and epigenetic regulation of gene expression. The prognostic analysis data show that the levels of CTCF are not directly associated with prognosis in different cancers (Voutsadakis, 2018). Conversely, amplifications of CTCFL are common in colorectal cancer, endometrial cancer, ovarian cancer, esophagogastric cancer, cervical cancer, non-small cell lung cancer, etc. CTCFL mRNA expression is not always associated with gene amplifications. The enrichment pathway is also focus on chromosome organization, mitotic cell cycle, DNA metabolic process and mRNA splicing. The prognostic analysis show that high CTCFL mRNA expressions are mainly associated with adverse prognosis in numerous cancers, except for in ER-positive breast cancer (Voutsadakis, 2018). The mutations, amplifications and the functional enrichment of CTCF and CTCFL in different cancers are illustrated in Fig. 2.

Figure 2 CTCF and CTCFL mutations and amplifications and functional enrichment in different cancers.

(A) The mutations and amplifications of CTCF and CTCFL in different cancers. The total number of samples examined was 2922 from ICGC/TCGA database in the cBioportal platform. (B) Genomic studies included in the cBioportal platform were examined of cases with CTCF and CTCFL mutations and co-occurring mutations. The functional enrichment of co-occurring mutations was analyzed with the Metascape platform.

The functions of CTCF and CTCFL

Initially, CTCF was identified as a transcriptional suppressor, in conjunction with the SP1 transcriptional factor, to suppress chicken c-myc gene transcription (Klenova et al., 1993; Lobanenkov et al., 1990). Subsequently, CTCF has been recognized for its role in interacting with cohesin complex to regulate long-range DNA interactions and form DNA interaction loops (Hansen, 2020; Liu, Wu & Wang, 2019). Cohesin, a circular protein complex composed of SMC3, SMC1 and RAD21 proteins, was originally identified in eukaryotic mitosis for its role in sister chromatid cohesin (Deng, Feng & Pauklin, 2022; Kojic et al., 2018; Remeseiro et al., 2012). Cohesin binds genomic sequences in a cis manner and extrudes DNA bidirectionally to form chromatin loops until it encounters boundaries preferentially bound by CTCF (Hansen, 2020). CTCF primarily functions as a chromatin insulator, restricting the action of regulatory elements to genes located within a TAD, thereby regulating transcriptional activation (Narendra et al., 2015; Tang et al., 2015). In acute myeloid leukemia (AML), HOTTIP mediated R-loop formation directly reinforces CTCF chromatin boundaries and regulates CTCF-mediated TAD integrity, driving β-catenin target gene expression and leukemia development (Luo et al., 2022). CTCF also plays a crucial role in the expression of uterine genes by mediating enhancer-promoter interactions during uterine development (Hewitt et al., 2023). When CTCF predominantly binds to intergenic or intronic regions, it acts more like an insulator, controlling gene expression in a more distal or indirect manner (Dehingia et al., 2022). As a chromatin insulator, CTCF demarcates the genome into active or inactive domains. Disruption of TAD boundaries can alter gene regulation by creating abnormal enhancer-promoter contacts, potentially leading to cancer.

Furthermore, CTCF can regulate cell death through MYC-p53-p21 signaling pathway (Pérez-Juste, Garcı a Silva & Aranda, 2000). It also influences cell senescence by blocking the expression of pericentromeric non-coding (ncRNA) transcribed from pericentromeric repetitive elements in young cells (Miyata et al., 2021). Consequently, CTCF can inhibit both cell death and cell senescence, thereby regulating genes associated with a variety of biological processes.

CTCF binding to DNA is inhibited by DNA methylation (Phillips & Corces, 2009), influencing processes such as gene imprinting (Bell & Felsenfeld, 2000), X chromosome inactivation (Chao et al., 2002) and exon alternative splicing (Shukla et al., 2011). At unmethylated CTCF binding sites, CTCF interacts with self PARylated PARP1 (poly [-ADP-ribose] polymerase 1) which prevents DNA methylation by inhibiting DNMT1 (DNA methyltransferase 1) activity (Zampieri et al., 2012). CTCF is implicated in numerous biological processes associated with higher-order chromatin structure and its binding to DNA is inhibited by DNA methylation. The functions of CTCF are shown in Fig. 3.

Figure 3 The function of CTCF in genome regulation.

(A) CTCF acts as transcriptional repressor to combine promoter and upstream silencer together. CTCF also acts as transcriptional activator to combine promoter and upstream enhancer together. (B) CTCF acts as an insulator by blocking the interaction between promoter and enhancer. (C) CTCF participates in gene imprinting, X chromosome inactivation and exon alternative splicing associating with DNA methylation. Igf2 is expressed from the paternal and H19 from the maternal allele. The ICR (imprinting control region) located in between the Igf2 and H19 genes. When ICR is methylated (Me) on the paternal allele, the methylation prevents CTCF binding. The enhancer located in the downstream of the H19 gene can interact with the Igf2 promoter and drive expression of Igf2. When ICR is nonmethylated on the maternal allele, derives ICR bound by CTCF, thereby preventing enhancer interaction with Igf2 promoter, resulting in an enhancer-promoter chromatin loop that enhances H19 expression. To achieve the mutually exclusive designation of active X (Xa) and inactive X (Xi), the process relies on the transcription of inactive X-specific transcript Xist, and Xist is blocked by the antisense gene Tsix. CTCF combines with unmethylated Xa chromosome to stimulate Tsix transcription or prevent Xist transcription. It leads to the inactivation of Xi chromosome. When the methylation of CTCF binding sites in Tsix, CTCF cannot combine with Tsix on the Xi chromosome. It leads to the transcription of Xist. When CTCF binding to alternative exon can promote inclusion of weak upstream exons by mediating local RNA polymerase II pausing. However, the binding is inhibited by DNA methylation resulting in faster movement of RNA polymerase II. It leads to increased skipping of alternative exon.

CTCFL is exclusively expressed in male germ cells, where waves of genome-wide demethylation created conditions conducive to CTCFL activation (Pugacheva et al., 2010; Renaud et al., 2007). It plays a crucial role in spermatogenesis by regulating expression of testis-specific genes. Male mice deficient in CTCFL exhibit significant subfertility due to meiotic defects. The absence of CTCFL impacts the expression of several testis-specific genes, including Gal3st1, Prss50 and Stra8 (Sleutels et al., 2012; Suzuki et al., 2010; Xu et al., 2004). These genes are integral to the processes of meiosis and sperm development. Thus, CTCFL has a profound influence on the transcriptional program governing spermatogenesis.

CTCFL also functions as a transcription factor in cancer, primarily binding to the promoters and enhancers of numerous genes. It occupies the promoters of key cancer-related genes such as c-MYC, hTERT, p53, INK/ARF, MDM2, PLK, PIM, suggesting its role in the deregulation of these genes during malignant transformation (Renaud et al., 2011). CTCFL expression is elevated in cancer stem cells (CSCs) (Soltanian & Dehghani, 2018), which are crucial for tumor growth, metastasis and treatment resistance. Silencing CTCFL leads to senescence and death of CSCs. CTCFL has been identified as an oncogene in various malignancies, including breast, prostate, ovarian cancer and colorectal carcinoma. In high-grade serous carcinoma (HGSC), CTCFL binds to the promoter of GALNT14 and enhances CTCF binding at numerous loci, and promotes cell motility and invasion (Hillman et al., 2019). In ALK-mutated, MYCN-amplified neuroblastoma cells that develop resistance to ALK inhibition, downregulation of MYCN expression is often accompanied by CTCFL overexpression, indicating that CTCFL may alter chromatin loops to support a resistance phenotype (Debruyne et al., 2019). Ectopic CTCFL expression in melanoma cells increases chromatin accessibility at the promoters of upregulated invasion-associated genes, indicating that CTCFL mediates a phenotypic switch from a proliferative to an invasive state (Moscona et al., 2023). CTCFL is also implicated in the activation of multiple cancer-testis genes, including PRAME, MAGEA1, MAGEA3, NY-ESO and others (Bhan et al., 2011; Hong et al., 2005; Vatolin et al., 2005). Known as CT27, CTCFL is classified as a cancer-testis antigen and has shown promise as a target for immunotherapy in cancer. Researchers successfully implemented CTCFL-targeting immunotherapeutic strategies in stringent 4T1 mouse and rat models of breast cancer (Loukinov, 2018; Loukinov et al., 2023), suggesting potential clinical applications that could improve the survival rates of breast cancer patients. In breast cancer cells, CTCFL binds to intronic regions, including the intronic alternative splicing exon 10 of the PKM (Pyruvate Kinase) gene, forming the cancer-specific PKM2 isoform that contributes to the Warburg Effect in breast cancer. Unlike CTCF mediated alternative splicing, CTCFL-dependent RNA polymerase II mediated alternative splicing of PKM2 depends on the methylation of exon 10 (Singh et al., 2017). Additionally, CTCFL can epigenetically reprogram clustered CTCF binding sites at intronic or intergenic regions into alternative transcriptional start sites, regulating cancer-testis genes alternative exons splicing in germ cells and CTCFL-positive cancers (Pugacheva et al., 2024). CTCFL plays a significant role in spermatogenesis and carcinogenesis, with numerous studies highlighting its transcriptional regulation of gene expression in sperm and cancers. However, the competitive binding of CTCFL to CTCF binding sites and its potential impact on disrupting 3D chromatin architecture remain poorly understood.

The collaboration of CTCF and CTCFL in spermatogenesis

The spermatogenesis is divided into three stages: (1) a small number of primitive type A spermatogonia (priSG-A) differentiate into type A spermatogonia (SG-A) and type B spermatogonia (SG-B) through mitosis. (2) type B spermatogonia (SG-B) forms spermatocytes (SC), which then produce haploid cells through two consecutive cell divisions (meiosis I and meiosis II). (3) After these cell divisions, spermatocytes form haploid round spermatids (rST) and spermatozoa (SZ) via spermiogenesis (Luo et al., 2020; Rooij, 2001). In normal cells, CTCF and CTCFL are co-expressed during all stages of spermatogenesis, particularly in meiotic and post-meiotic round spermatids, suggesting that both proteins competitively bind to some genomic target sites at 2 ×CTSes with enrichment of RNA pol II, H3K4me3 and H3K27ac (Lobanenkov & Zentner, 2017). Although CTCF and CTCFL co-locate in preleptotene spermatocytes and round spermatids, CTCFL is functionally distinct from CTCF, and they are not interchangeable. CTCF-only sites function in chromatin architecture. CTCF is expressed throughout spermatogenesis and is required for the maintenance of synaptonemal structure and homologous recombination (Hernández-Hernández et al., 2016; Wang et al., 2019). CTCFL preferentially acts as a transcriptional regulator to regulate the testis-specific gene expression. Recent studies have shown that CTCF&CTCFL and CTCFL-only sites are occupied by several promoters of testis-specific transcriptional regulators (TSTRs) which regulate testis-specific genes (Rivero-Hinojosa et al., 2017; Sleutels et al., 2012). Due to the differences in 3D structures, CTCFL is unable to bind cohesin, while the meiotic cohesins (e.g., REC8 complex and RAD21L complex) localize to the CTCFL binding sites, rather than CTCF sites (Boukaba et al., 2022). CTCF and cohesin remain associated throughout meiosis, suggesting that disruption and reorganization of TADs depend on CTCFL expression rather than the clearance of CTCF and cohesin (Luo et al., 2020). CTCFL overexpression can induce rapid changes in the expression of testis-specific genes through the disruption of TADs either before or during the meiosis. CTCFL is likely to interact with meiosis-specific cohesin complexs to establish novel CTCFL dependent chromatin loops, thereby contributing to the genome-wide organization observed in round spermatids (Lobanenkov & Zentner, 2017; Luo et al., 2020). Consequently, CTCFL plays an essential role in chromatin remodeling and the organization of the sperm genome during the spermatogenesis (Pugacheva et al., 2015). The dynamic reorganization of chromatin loops and TADs throughout spermatogenesis is modulated by CTCFL expression.

CTCF and CTCFL are co-expressed throughout all stages of spermatogenesis and bind to overlapping target sites. Their functional interplay in the testis is significant for the development of germ cells. The mice that are null for ctcf result in the early embryonic lethality due to the specific function of CTCF (Moore et al., 2012). While CTCF alone is insufficient for the regulation of the testis-specific genes in germ cells, the collaboration between CTCF and CTCFL is essential. Moreover, CTCFL is necessary for the normal expression of CTCF in male germ cells. Although CTCFL knockout mice exhibit a fertile phenotype with only minimal germ cell defects, including delayed production of round spermatids, the presence of CTCF partially compensates for the absence of CTCFL, maintaining the functionality of CTCF-binding sites. However, compound mutant mice, which are heterozygous for Ctcf and homozygous for Ctcfl knockout, display a complete infertility phenotype accompanied by aberrant meiotic recombination (Rivero-Hinojosa et al., 2021). The simultaneous deletion of CTCF and CTCFL results in severe deregulation of spermatogenesis-specific genes, such as Tsp50 and Gal3st1, which are directly regulated by CTCFL and essential for meiosis (Kosaka-Suzuki et al., 2011; Sleutels et al., 2012; Suzuki et al., 2010; Xu et al., 2004). Therefore, knockout of either CTCF or CTCFL only slightly impaired testis-specific gene expression, but simultaneous deletion of CTCF and CTCFL lead to the dramatic deregulation of these target genes. During the spermatogenesis, CTCFL disrupts the chromatin loops established by CTCF through binding meiotic cohesins, allowing the expression of testis-specific genes and re-establishing novel chromatin loops and TADs. CTCF can partially compensate the effect for the absence of CTCFL, but it cannot completely replace the function of CTCFL.

The interference of CTCF and CTCFL in carcinogenesis

CTCF and cohesin are essential for maintaining chromatin interactions and organizing TADs. CTCF can transcriptionally suppress CTCFL expression in somatic cells. Disruption of TAD boundaries mediated by CTCF and cohesin can result in abnormal enhancer-promoter contacts, which are implicated in cancer development. Most CTCFL-regulated genes are little overlap with those of CTCF. As mentioned above, CTCFL is predominantly located at active promoters and enhancers rather than CTCF located at intronic or intergenic regions (Nishana et al., 2020). When CTCF is located at promoters preferentially occurs at CTCF+CTCFL overlapping sites. Although expression of CTCFL in the presence of CTCF does not dramatically alter TADs at a global level, ectopic CTCFL expression disrupts CTCF-mediated TADs and impacts on expression of genes within altered loops at CTCF+CTCFL overlapping sites (Nishana et al., 2020). Cancer cells often possess the characteristics similar to those in male germ cells, including the expression of testis-specific genes. In normal somatic cells, CTCF transcriptionally represses alternative cancer-testis-specific transcription at intergenic and intronic CTCF binding sites. However, in cancer cells, CTCFL likely competes with CTCF for binding to DNA target sites. These CTCF binding sites can be epigenetically reprogrammed into active alternative promoters by CTCFL, leading to the activation of cancer-testis-specific genes. As a pioneer transcription factor, CTCFL firstly recruits the chromatin remodeling factor, SRCAP, which facilitates the replacement of H2A histone with H2A.Z. This process results in an open chromatin conformation and then facilitates the recruitment of additional transcription factors, thereby activating transcription sites (Pugacheva et al., 2024). Importantly, overexpression of CTCFL disrupts CTCF-mediated chromatin loops, increasing CTCF occupancy and leading to the establishment of new chromatin loops and the formation of super-enhancers. These alterations drive the ectopic long-distance gene expression associated with cell cycle and inflammation and facilitate the cancer development (Nishana et al., 2020; Pugacheva et al., 2024). Moreover, CTCFL may establish novel chromatin loops to support cancer-specific resistance phenotypes (Debruyne et al., 2019), while CTCFL cannot rescue TADs in the absence of CTCF. When CTCFL expresses in the absence of CTCF, cohesin peaks are globally depleted. CTCFL is unable to act as insulator like CTCF. Thus, CTCFL’s competitive binding to CTCF DNA binding sites can result in aberrant gene expression and have a widespread impact on TAD structure. The distinct mechanisms by which CTCF and CTCFL regulate target genes are illustrated in Fig. 4.

Figure 4 The pattern of CTCF and CTCFL in genome regulation through chromatin architecture.

(A) CTCF and cohesin are responsible for the establishment of the 3D organization of the genome in normal somatic cells. (B) When CTCF degraded, low level of CTCF is not sufficient to form stable homodimers at the 2×CTSes but may be sufficient for CTCF monomers to occupy the 1×CTSes. Upon such a loss of CTCF at the 2×CTSes, disruption of CTCF-mediated chromatin loops leads to unexpectedly dramatic deregulation of target genes even lethality. (C) When CTCF and CTCFL are co-expressed in germline, CTCF and CTCFL at 2×CTSes are cooperatively and competitively involved in transcriptional activation of testis-specfic genes to regulate the chromatin loops in testis. CTCFL binding recruits epigenetic reprogramming, which replaces H2A histone with H2A.Z, leading to the opening of chromatin. (D) When CTCF degraded and CTCFL overexpressed, CTCFL occupys at the 2×CTSes and disrupts CTCF-mediated chromatin loops and then activates multiple abnormal transcripts in cancers.

Conclusions and perspective

In summary, CTCF and cohesin are critical for establishing the 3D organization of the genome in normal somatic cells. Degradation of CTCF disrupts CTCF-mediated chromatin loops, leading to significant deregulation of target genes and potential lethality. In the testis, which exhibits unique chromatin dynamics, loops and TADs are actively reorganized. CTCFL plays a key role in chromatin remodeling and sperm genome organization, functioning in conjunction with CTCF. During germline development, both CTCF and CTCFL are co-expressed and bind to specific genomic regions, termed 2 ×CTSes (both CTCF and CTCFL occupy sites). These 2 ×CTSes are predominantly associated with active promoters and enhancers in germ cells. CTCFL facilitates epigenetic reprogramming by replacing H2A histone with H2A.Z, leading to the chromatin opening and subsequent recruitment of additional transcription factors. CTCF and CTCFL at 2 ×CTSes interact cooperatively and competitively to regulate the dynamic transcriptional activation of testis-specfic genes and regulate the chromatin loops. Overexpression of CTCFL in cancer cells leads to genomic instability and activation of aberrant transcripts, thereby altering chromatin loop structures.

Re-expression of CTCFL is associated with genome-wide DNA hypomethylation in both sperm and cancer cells. Despite this, CTCFL can bind to methylated DNA, and the precise interactions between CTCFL and DNA methyltransferase remain inadequately understood. Furthermore, the elevated expression of CTCFL is not a universal feature across all cancers, raising questions about the factors influencing its differential expression in cancer cells. It is also unclear whether ectopic CTCFL expression and CTCF degradation affect CTCF binding sites and subsequently regulate target genes in other disease contexts. Future research should focus on elucidating the mechanisms by which aberrant CTCFL expression disrupts CTCF-mediated chromatin loops and impacts epigenetic regulation.

We thank Dr. Ashiq Ali for English language editing.

Additional Information and Declarations

Competing Interests

Author Contributions

Data Availability

The authors declare there are no competing interests.

Xin Tong conceived and designed the experiments, performed the experiments, analyzed the data, prepared figures and/or tables, authored or reviewed drafts of the article, and approved the final draft.

Yang Gao analyzed the data, authored or reviewed drafts of the article, and approved the final draft.

Zhongjing Su conceived and designed the experiments, performed the experiments, analyzed the data, prepared figures and/or tables, authored or reviewed drafts of the article, and approved the final draft.

The following information was supplied regarding data availability:

This is a literature review.

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
