# Peer review of "Interaction of CTCF and CTCFL in genome regulation through chromatin architecture during the spermatogenesis and carcinogenesis"

_PeerJ, doi:10.7717/peerj.18240_

## Round 0.1 · original submission · Major Revisions

Please carefully address all the issues pointed out by the reviewers and revise your manuscript accordingly.

**Language Note:** The review process has identified that the English language must be improved. PeerJ can provide language editing services - please contact us at [email protected] for pricing (be sure to provide your manuscript number and title). Alternatively, you should make your own arrangements to improve the language quality and provide details in your response letter. – PeerJ Staff

Reviewer 1 ·

Basic reporting

The manuscript titled “Interaction of CTCF and CTCFL in genome regulation through chromatin architecture during spermatogenesis and carcinogenesis” provides a detailed review of the roles of CTCF and CTCFL proteins in chromatin architecture and gene regulation. It particularly focuses on their functions during spermatogenesis and in various cancer contexts. The conclusions are appropriate and supported by the cited literature.

Experimental design

The literature survey predominantly focuses on recent PubMed articles from the past five years. While this ensures the inclusion of up-to-date research, it might overlook crucial historical perspectives. While I am glad to see the authors included the initial discovery of CTCFL and CTCF. I think the author could also include some earlier studies which introduce, the structural characterization of CTCF and CTCFL, how they interact each other and with cohesin, and their similarity and difference in function and expression patterns. These are important for a thorough understanding for readers.

Validity of the findings

Regarding the figures, particularly Figure 1 and Figure 2A, which display extensive sequence alignments, they appear overwhelming, and provide limited additional insight. Such alignments are readily producible using standard bioinformatics software using sequences available in previous literatures. I feel these figures could be simplified in favor of more interpretive visualizations that add meaningful value to the review.

Reviewer 2 ·

Basic reporting

1. Language and Structure:
The manuscript is written in clear, professional English, adhering to the standards expected for scientific literature. The structure conforms to the norms of PeerJ, facilitating easy navigation and comprehension. The introduction and background sections effectively introduce the subject, providing sufficient context and motivation for the study.

2. Literature Referencing:
The literature review is comprehensive, covering relevant studies that inform the discussion on CTCF and CTCFL. The references are pertinent and recent, with key historical studies included to give a thorough background.

3. Figures and Images:
The figures are appropriately utilized to illustrate complex concepts and data. However, there should be a check for any inappropriate manipulations of images. Clearer legends and more detailed descriptions would enhance their effectiveness.

Experimental design

1. Scope and Aims:
The article fits within the aims and scope of PeerJ, presenting a rigorous review of the literature on CTCF and CTCFL in genome regulation. The investigation is conducted to a high technical and ethical standard, with methodologies described in sufficient detail to allow replication.

2. Methodology:
The survey methodology is consistent with a comprehensive, unbiased coverage of the subject. However, the review would benefit from a more detailed explanation of the criteria used for selecting studies and the inclusion of a PRISMA flow diagram to illustrate the literature selection process.

Validity of the findings

1. Protein Interaction Networks:
The manuscript provides a detailed examination of CTCF and CTCFL roles in chromatin architecture and gene regulation. However, it lacks an in-depth discussion on the network of proteins that interact with CTCF and CTCFL and how these interactions affect cellular phenotypes and human diseases. Expanding on this aspect would significantly enhance the manuscript's comprehensiveness.

2. Co-occurring Mutations and Pathways:
While the authors outline the prevalence of CTCF mutations in various tumors, they do not present data on co-occurring mutations or the enrichment of other pathways in these patients. Including such data would provide a more complete picture of the genetic landscape in which CTCF operates and its implications for cancer biology.

3. Prognostic Impact:
The review mentions the prevalence of CTCF and CTCFL mutations but fails to discuss their prognostic effects. Presenting data on the prognostic significance of these mutations in different cancers would add valuable clinical insight and relevance to the study.

Additional comments

The review provides a solid foundation on the roles of CTCF and CTCFL in genome regulation. However, to elevate the manuscript, the authors should:

Expand on the network of proteins interacting with CTCF and CTCFL and their impact on cellular phenotypes and diseases.
Present data on co-occurring mutations and other enriched pathways in patients with CTCF mutations.
Discuss the prognostic effects of CTCF and CTCFL mutations in various cancers.

·

Basic reporting

The manuscript by Tong et al. reviews recent literature on the role of the two paralogous proteins, CTCF and CTCFL (BORIS), in the 3D genome organization of germ and cancer cells, where both genes are co-expressed. Cell-specific gene transcription depends on 3D genome organization, which is mediated by CTCF and cohesin in all vertebrates. While the functional role of CTCF has been well addressed in thousands of publications, the role of CTCFL is less clear, though it should be related to CTCF function as both paralogous proteins share DNA target sites. Thus, the goal of this review is to address how CTCF and CTCFL share and compete for the target sites during spermatogenesis and carcinogenesis.
Unfortunately, I cannot recommend the manuscript in its current state for publication. The manuscript requires major revisions and significant English corrections. The primary issue is that the context is difficult to follow, making it challenging to understand the underlying ideas. The text is presented in a disorganized manner, lacking logical flow.
I recommend that the authors divide the review into smaller chapters, each addressing a specific question, to provide a clear flow on how CTCFL expression affects 3D genome organization mediated by CTCF. Based on their text, the chapters could address: CTCF and CTCFL expression, their protein structures, their functions, their target sites, their collaboration in spermatogenesis, their interference during carcinogenesis, and so on.
There are many reviews addressing the multifunctionality of CTCF. I would recommend the authors focus on the cooperation and competition between CTCF and CTCFL in respect to 3D genome regulation, instead of listing CTCF functions.
Major Concerns (only some examples listed):
1. Introduction: The text in the Introduction is difficult to follow, consisting of a list of facts that are not connected. This needs restructuring for better coherence.
2. English corrections: The manuscript requires extensive English corrections. It is challenging to understand what the authors are trying to convey. There is a list of conclusions from different studies that are not connected. For example, lines 114-118 describe CTCF binding motifs and the next lines 118-122 suddenly state that the CTCF N-terminus is necessary for cohesin retention.
3. Specific line issues (just some examples): Line 17: The phrase "ubiquitously expressed" means expressed in all types of tissues, making "in multiple tissues" redundant; Line 24: Should be "binds to CTCF target sites."; Line 25: "…how to change the chromatin architecture will be discussed" is not correct English.
4. There are also sentences that do not make much sense scientifically, not just due to English errors. Here are some examples:
• Line 61-62: "The nucleosome is the first order of chromatin structure, which DNA is surrounded by histone into octamer" should be "The nucleosome is the first level of chromatin structure, where DNA is wrapped around a histone octamer."
• Line 170-171: "CTCF and CTCFL are co-expressed in testis and some cancer cells at 2×CTSes" is unclear.
• Line 192-195: The description of cohesin and its components is confusing and needs clarification.
• Line 224-225: "CTCF also can actively inhibit DNA methylation at CTCF binding sites in gene imprinting" needs rephrasing for clarity.
• Line 340-342: The sentence lacks clarity.

Experimental design

no comment

Validity of the findings

no comment

Additional comments

no comment

---

## Round 0.2 · Minor Revisions

Please address the remaining concerns of the reviewers and revise the manuscript accordingly.

Reviewer 1 ·

Basic reporting

no comment

Experimental design

no comment

Validity of the findings

no comment

Additional comments

I am happy to see that the authors addressed my previous comments on Fig1 and Fig2, added earlier studies which introduced the discovery of CTCF and CTCFL, the structural characterization of CTCF and CTCFL etc.

To further enhance the clarity of Figure 1A, I suggest adding the 3D structures of both proteins, with the zinc finger (ZF) regions and other important motifs highlighted on the structures. (also showing the binding sites overlap). This is more useful than showing the sequence alignment, as these structure differences are important for understanding the functional differences between CTCF and CTCFL.

Reviewer 2 ·

Basic reporting

The Authors resubmit a revised version of their comprehensive review.
They have addressed all my requests and the review is vastly improved.
I suggest that the paper can be accepted at the current form.

Experimental design

See above

Validity of the findings

See above

Additional comments

See above

·

Basic reporting

The manuscript has improved, but there is still a lot of repetitive information. I have some minor corrections:

1. Line 97: It was ChIP-seq, not ChIP-chip.
2. Line 116: The authors cite Xiao et al. (PMID: 21444719) from 2011 but list the year as 2023 (Lines 643-645).
3. Lines 128-130: CTCFL isoforms are described in PMID: 21079786, not in Fugmann et al. (454-457). There is no such publication as Fugmann et al.
4. Line 260: It should be Prss50, not Prss 50.
5. Line 261: The reference Xu et al., 2004 is not relevant to CTCFL regulation of Prss50.
6. Line 315: The correct reference should be PMID: 28145452, not Sleutels, 2012.
7. Line 336: The reference for null mice is missing.

Experimental design

no comments

Validity of the findings

no comments

Additional comments

no comments

---

## Round 0.3 · accepted · Accept

All remaining issues indicated by the reviewers were successfully addressed and the revised manuscript is acceptable now.